# OCRbayes: A Bayesian hierarchical modeling framework for Seahorse extracellular flux oxygen consumption rate data analysis

Xiang Zhang[1,2], Taolin Yuan[2], Jaap Keijer[2], Vincent C. J. de Boer[2]*

**1** Theoretical Biology and Bioinformatics, Utrecht University, Utrecht, The Netherlands, **2** Human and Animal Physiology, Wageningen University, Wageningen, The Netherlands

* vincent.deboer@wur.nl

## Abstract

### Background

Mitochondrial dysfunction is involved in many complex diseases. Efficient and accurate evaluation of mitochondrial functionality is crucial for understanding pathology as well as facilitating novel therapeutic developments. As a popular platform, Seahorse extracellular flux (XF) analyzer is widely used for measuring mitochondrial oxygen consumption rate (OCR) in living cells. A hidden feature of Seahorse XF OCR data is that it has a complex data structure, caused by nesting and crossing between measurement cycles, wells and plates. Surprisingly, statistical analysis of Seahorse XF data has not received sufficient attention, and current methods completely ignore the complex data structure, impairing the robustness of statistical inference.

### Results

To rigorously incorporate the complex structure into data analysis, here we developed a Bayesian hierarchical modeling framework, OCRbayes, and demonstrated its applicability based on analysis of published data sets.

### Conclusions

We showed that OCRbayes can analyze Seahorse XF OCR experimental data derived from either single or multiple plates. Moreover, OCRbayes has potential to be used for diagnosing patients with mitochondrial diseases.

## Introduction

Mitochondria are double-membrane organelles that are central hubs in regulating energy generation and partitioning. Patients with genetic defects in mitochondrial function are often affected by severe and progressive disease in early life [1]. Furthermore, mitochondrial disorders have also been found to be involved in cardiovascular diseases [2], type II diabetes [3],

---

**Data Availability Statement:** The datasets analysed during the current study are available in the Github repository (https://github.com/XiangZhangSC/seahorse).

**Funding:** The author(s) received no specific funding for this work.

**Competing interests:** The authors have declared that no competing interests exist.

neurodegenerative disease [4] and cancer [5]. Thus, restoring mitochondrial function is emerging to be a therapeutic target for both common diseases [6] as well as genetic mitochondrial diseases [7].

Mitochondria produce energy in oxidative phosphorylation (OXPHOS) primarily by transferring electrons along the electron transport chain (ETC) on the inner membrane. Along the ETC, there are four complexes (complex I, II, III and IV), together building up a proton gradient that is ultimately used by ATP synthase to generate ATP. Since electrons are primarily accepted by $O_2$ to produce $H_2O$, the OXPHOS activity can be assessed by measuring oxygen consumption rate (OCR). As a reliable and efficient platform, Seahorse XF analyzer provides a multiwell plate based respirometry assay that is widely used to quantify OCR in living cells [8]. Typically, the Seahorse XF analyzer measures the OCR of cells in a 96-well plate under different ETC/OXPHOS perturbation scenarios, used for assessing mitochondrial functionality such as maximal respiration, leak respiration and ATP-linked respiration [9].

A typical Seahorse assay includes three measurement cycles for each phase. Every measurement cycle starts by lowering the cartridge and creating a temporary semi-closed ~2 $\mu$L chamber [10]. During a measurement cycle, fluorescent oxygen sensors capture oxygen concentration changes in the chamber and outputs OCR. The Seahorse XF analyzer measures OCR in tens to hundreds of thousands of cells per well and typically requires 4–5 replicate wells per experimental group, allowing the analysis of multiple experimental groups in one plate [11]. Since measurement cycles are nested within phases, phases are crossed with wells, and wells are nested in experimental groups, a very complex structure is inherently embedded in the Seahorse XF OCR data. Surprisingly, current Seahorse XF data analyses most often ignore this complex structure, and by default, data sets are often chopped into subgroups followed by performing ANOVA-like statistical tests. Although there are advanced tools developed for Seahorse data analysis [12, 13], none of them comprehensively take the complexity of the data structure into account. As a result, OCR variation between measurement cycles, replicate wells and replicate plates is overlooked, impairing the robustness of the interpretation of Seahorse XF OCR outcomes, and eventually the development of mitochondrial targeted therapies as well as our understanding of mitochondrial biology.

A natural way to incorporate the complex structure into data analysis is to use hierarchical modeling, which has been extensively developed for analyzing gene expression data [14, 15]. Here we developed a Bayesian hierarchical modeling framework, OCRbayes, for the Seahorse XF OCR data analysis. Compared to the currently most advanced Seahorse data analysis tool, OCR-stats which is a frequentist approach, OCRbayes is based on a fully Bayesian approach due to its flexibility and convenience for constructing the hierarchical models. To demonstrate the applicability of our approach as well as its potential implication for mitochondrial disease diagnostics, we applied OCRbayes to a publicly available OCR data set [13], which contains over 200 Seahorse experiments performed on human fibroblasts derived from patients with mitochondrial diseases and controls.

## Method

### OCRbayes: A Bayesian hierarchical modeling framework for Seahorse OCR data analysis

In order to incorporate the complex data structure into the analysis of Seahorse XF OCR measurements, we developed a Bayesian hierarchical modeling framework, OCRbayes. In this study, we focused on experimental data containing two groups, such as patient and control group. From our perspective, the Seahorse OCR data include three levels, including 1) measurement cycle, 2) well and 3) plate.

During a measurement cycle, the Seahorse XF analyzer uses fluorescent oxygen sensors to track OCR. For every interval, multiple measurement cycles are performed in order to accurately measure the OCR. A typical Seahorse assay contains four intervals. The first interval refers to initial phase, and the second, third, fourth interval refer to phase after injecting oligomycin (blocking proton translocation through ATP synthase), FCCP (allowing protons to move into the mitochondrial matrix independent of the ATP synthase) and antimycin/rotenone (inhibiting complex I and III, shutting down mitochondrial respiration).

The variation between the OCR values within an interval is the between measurement cycle variation. Since OCR values must be positive, we used a lognormal distribution with the true OCR value at the log scale ($logOCR_{true}$) and the between measurement cycle standard deviation (also at the log scale, $logOCR_{sd}$) to model each observed OCR value ($OCR_{obs}$).

$$OCR_{obs}[i] \sim Lognormal(logOCR_{true}[P[i], W[i], I[i]], logOCR_{se}[P[i], I[i]]) \qquad (1)$$

$OCR_{obs}$ is a vector of length $N_{plate} \times N_{well} \times N_{interval} \times N_{measurement}$, where $N_{plate}$, $N_{well}$, $N_{interval}$ and $N_{measurement}$ are number of plates, wells, intervals and measurement cycles, respectively. The $logOCR_{true}$ refers to $N_{plate}$ layers of matrices. Each layer is a matrix with $N_{well}$ rows and $N_{interval}$ columns. $logOCR_{sd}$ is a matrix of $N_{plate}$ rows and $N_{interval}$ columns.

In a typical Seahorse XF assay, one cell line undergoing the same experimental treatment is seeded in multiple wells. However, the OCRs in these replicate wells will not be the same, and the difference is called between well variation. One obvious reason causing the between well variation is that the number of cells in these replicate wells are not identical. To adjust for the effect of cell number difference, we modeled the true OCR value as a function of the cell number, and from there estimated $OCR_{per\ 1k\ cells}$, which represents OCR value per 1000 cells that received the same treatment (or from the same group) and injection. In addition to cell number difference, technical, procedural or instrumental noise can also contribute to between well variation. We captured this well-to-well variation after accounting for cell number difference with the residual parameter $\sigma_{well}$.

$$\begin{aligned} logOCR_{true}&[P[i], W[i], I[i]] \\ &\sim Normal(OCR_{per\ 1k\ cells}[P[i], G[i], I[i]] \times N_{cells}[W[i]], \sigma_{well}[P[i], G[i], I[i]]) \end{aligned} \qquad (2)$$

$OCR_{per\ 1k\ cells}$ is a three-dimensional matrix with $N_{plate}$ layers, and each layer has $N_{group}$ rows and $N_{Interval}$ columns. $N_{cells}$ is a vector of length $N_{well}$, and every entry represents the cell number in that well.

Apart from the technical replicates in one plate, the biological insight for a (patient) cell line or a specific condition is generally validated by repeating the Seahorse assay on different days. As a result, OCR data are distributed on more than one plate. Due to batch effects such as plating, culturing or environmental differences between time and laboratories, OCR measurements will differ between plates. To take into account the between plate variation, we used another lognormal distribution with the logarithm transformed OCR value per 1000 cells, ($\mu_{OCR_{per\ 1k\ cells}}$) and the between plate standard deviation ($\sigma_{plate}$) to model the OCR value per 1000 cells $OCR_{per\ 1k\ cells}$.

$$OCR_{per\ 1k\ cells}[P[i], G[i], I[i]] \sim Lognormal(log(\mu_{OCR_{per\ 1k\ cells}}[G[i],\ I[i]]), \sigma_{plate}[G[i],\ I[i]]) \qquad (3)$$

## Bayesian inference

OCRbayes focuses on calculating posterior distributions for OCR per 1000 cells in various experimental conditions ($P(\mu_{OCR_{per\ 1k\ cells}}|OCR_{obs})$). The posterior distributions were combinations of prior distributions ($P(logOCR_{se})$, $P(\sigma_{well})$, $P(\sigma_{plate})$ and $P(\mu_{OCR_{per\ 1k\ cells}})$), and the likelihood function ($P(OCR_{obs}|N_{cell}, \mu_{OCR_{per\ 1k\ cells}}, logOCR_{sd}, \sigma_{well}, \sigma_{plate})$)

In this study, since our case studies focused on human fibroblast cells, we used informative prior distributions for $logOCR_{sd}$, $\sigma_{well}$, $\sigma_{plate}$ and $\mu_{OCR_{per\ 1k\ cells}}$. Since values of all these parameters must be positive, we used four lognormal distributions. We applied maximum likelihood estimation to calculate the lognormal distribution parameters based on the Seahorse XF OCR data in OCR-stats [13] by running the fitdistr function built in the MASS r package. In this study, our prior distributions are

$$
\begin{aligned}
logOCR_{sd} &\sim Lognormal(-3.23,\ 0.79) \\
\sigma_{well} &\sim Lognormal(-1.62,\ 0.53) \\
\sigma_{plate} &\sim Lognormal(-1.18,\ 0.05) \\
\mu_{OCR_{per\ 1k\ cells}} &\sim Lognormal(0.3,\ 0.79)
\end{aligned}
$$

The Bayesian multi-level model of OCRbayes was implemented in Stan (version 2.19.3) [16]. We fitted the model by running Hamiltonian Markov Chain Monte Carlo. We ran four Markov chains with 2000 iterations in each chain. The code can be found at https://github.com/XiangZhangSC/seahorse. We also provided the R code of OCRbayes in the supplementary file.

## Calculation of bioenergetic measures

Based on OCR (per 1000 cells), we calculated various bioenergetic measures, such as basal respiration, proton leak, ATP-linked respiration, spare respiratory capacity and maximal respiration. These bioenergetic measures are defined as below.

$$
\text{Basal respiration} = OCR_{per\ 1k\ cells,\ initial} - OCR_{per\ 1k\ cells,\ antimycin/rotenone} \tag{4}
$$

$$
\text{ATP-linked respiration} = OCR_{per\ 1k\ cells,\ initial} - OCR_{per\ 1k\ cells,\ oligomycin} \tag{5}
$$

$$
\text{Proton leak} = OCR_{per\ 1k\ cells,\ oligomycin} - OCR_{per\ 1k\ cells,\ antimycin/rotenone} \tag{6}
$$

$$
\text{Spare respiratory capacity} = OCR_{per\ 1k\ cells,\ FCCP} - OCR_{per\ 1k\ cells,\ initial} \tag{7}
$$

$$
\text{Maximal respiration} = OCR_{per\ 1k\ cells,\ FCCP} - OCR_{per\ 1k\ cells,\ antimycin/rotenone} \tag{8}
$$

## Human fibroblast OCR data

To benchmark OCRbayes as well as illustrate how OCRbayes can be used for analyzing OCR in patients with mitochondrial diseases, we used the OCR data set provided in Yepez et al. [13]. This data set contains Seahorse OCR measurements from 203 human fibroblast cell lines that have been assayed in 126 plates. Normal human dermal fibroblast (NHDF) reference cell lines were used as controls in all plates. The other 202 cell lines were derived from patients suspected to suffer from rare mitochondrial diseases. Among these 202 cell lines, 26 fibroblast cell lines were measured in multiple plates. We used the 176 patient fibroblast cell lines that were

assayed in a single plate as well as the control cell line in the same plates for estimating prior distributions. The other 26 fibroblast cell lines were used for benchmarking OCRbayes. Among the 202 cell lines that were analyzed, Yepez et al. [13] labeled 6 patient cell lines as positive controls that have shown statistically significant reduction in maximum respiration. Meanwhile, Yepez et al. also labeled another two patient cell lines as negative controls, since these cell lines did not show changes in OCR in earlier experiments [13].

We processed the original data by removing wells in which single or more OCR measurements were missing. After filtering, we used 176 patient cell lines together with the NHDF control cell line on 78 plates for estimating the prior distributions for between measurement cycle variation (logOCR$_{sd}$) and between well variation ($\sigma_{well}$). To estimate the prior distribution for between plate variation ($\sigma_{plate}$) and mean OCR per 1000 cells ($\mu_{OCR_{per\ 1k\ cells}}$), we used the OCR values from the NHDF cell lines that were plated in all 78 plates.

## Statistical analysis

For benchmarking OCRbayes, we compared the patient cell lines to the control cell line (NHDF). Since the maximal respiration was reported as the primary mitochondrial dysfunction outcome in the Yepez et al. study [13], we reported here the mean log2 fold change of maximal respiration together with the False Discovery Rate (FDR). FDR was calculated based on the posterior error probability $(1 - P(\log_2\left(\frac{patient}{control}\right)|OCR_{obs}) < 0))$, where $P(\log_2\left(\frac{patient}{control}\right)|OCR_{obs})$ represented the posterior distribution of fold change after fitting the Bayesian model to the experimental OCR data. If the FDR was below 0.05, we considered that the difference between patient and control cell line was statistically significant.

## Results

### OCRbayes: From OCR to respiration metric difference

To demonstrate the applicability, we applied OCRbayes to analyze OCR data derived from two patient cell lines with known mutations in either *BOLA3* or *PET100* gene. These two genes encode proteins that are essential for biogenesis or assembly of mitochondrial complexes [17–19]. We compared the patient cell line to the control cell lines within their respective plates. These two cell lines were both assayed on two plates on two different days. For both cell lines, we observed a clear decrease in maximal respiration compared to the control cell lines in both plates (Fig 1A and 1E). Meanwhile, we noticed that the range of OCR values in the two plates used for profiling *BOLA3* patient cell line differed considerably (Fig 1A). In particular, maximal OCR value in the first plate was around 200 pmol/min, whereas in the second plate the maximal OCR value was around 100 pmol/min. In contrast, the range of OCR values in the two plates used for profiling *PET100* were similar to each other. In addition, it was obvious that there was considerable variation between the replicate wells in both plates for both cell lines.

By applying OCRbayes, we combined the two plates for each cell line, calculated the posterior distributions for OCR per 1000 cells during the four intervals of a typical Seahorse assay, including initial phase without injection (Int1), oligomycin phase (Int2), FCCP phase (Int3) and antimycin/rotenone phase (Int4) (Fig 1B and 1F). Next, the calculated posterior OCR per 1000 cells for the four intervals were transformed into the various respiration metrics, including ATP-linked respiration, basal respiration, maximal respiration, proton leak and spare respiratory capacity (Fig 1C and 1G). In the last step, we compared the respiration metrics in patient cell line to the control cell line, and calculated the posterior distribution for log2(fold change) (Fig 1D and 1H). We observed that *BOLA3* patient cell line showed reduction in basal

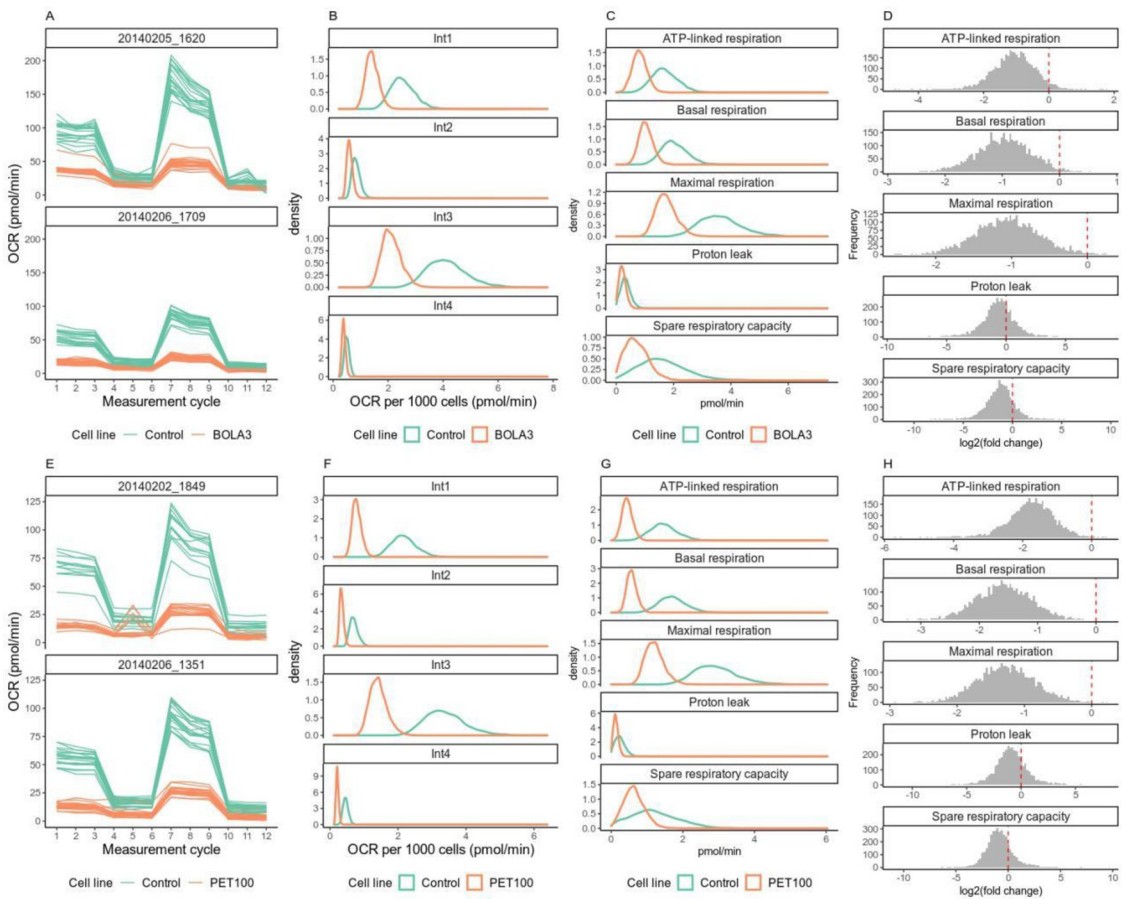

**Fig 1.** (A) and (E) are OCR profiles for patient cell lines that have a genetic mutation in either BOLA3 (A) or PET100 (E), compared to control cell lines in two plates. (B) and (F) are posterior distributions for OCR per 1000 cells in BOLA3 (B) and PET100 (F) mutated patient cell lines during initial phase (Int1), oligomycin phase (Int2), FCCP phase (Int3) and antimycin/rotenone phase (Int4). (C) and (G) are posterior distributions for respiration metrics in BOLA3 (C) and PET100 (G) mutated patient cell lines and control cell lines. (D) and (H) are posterior distributions of log2 fold change in the respiration metrics between patient and control cell line.

respiration (posterior mean log2(fold change) -0.974, 95% credible interval [-1.93, -0.0970]) and maximal respiration (-1.06 [-1.87,-0.228]), compared to the control cell line (Fig 1D). On the other hand, we observed no difference in ATP-linked respiration (-1.08 [-2.38, 0.117]), proton leak (-0.630 [-3.87, 2.32]) and spare respiratory capacity (-1.26 [-4.41,1.62]) between *BOLA3* patient cell line and the control cell line (Fig 1D). Meanwhile we observed lower ATP-basal respiration (-1.75 [-3.18, -0.546]), basal respiration (-1.58 [-2.52, -0.667]) and maximal respiration (-1.28 [-2.10, -0.436]) in the *PET100* patient cell line than the control cell line (Fig 1H). We observed no difference in proton leak (-0.906 [-4.14,2.54]) and spare respiratory capacity (-0.826 [-3.29, 2.34]) in the *PET100* patient cell line compared to the control cell line (Fig 1H).

## OCRbayes accounts for various technical variations during Seahorse XF OCR data analysis

OCR measurements generated by the Seahorse XF analyzer were affected by various technical variations, including between measurement cycle variation, between well variation and

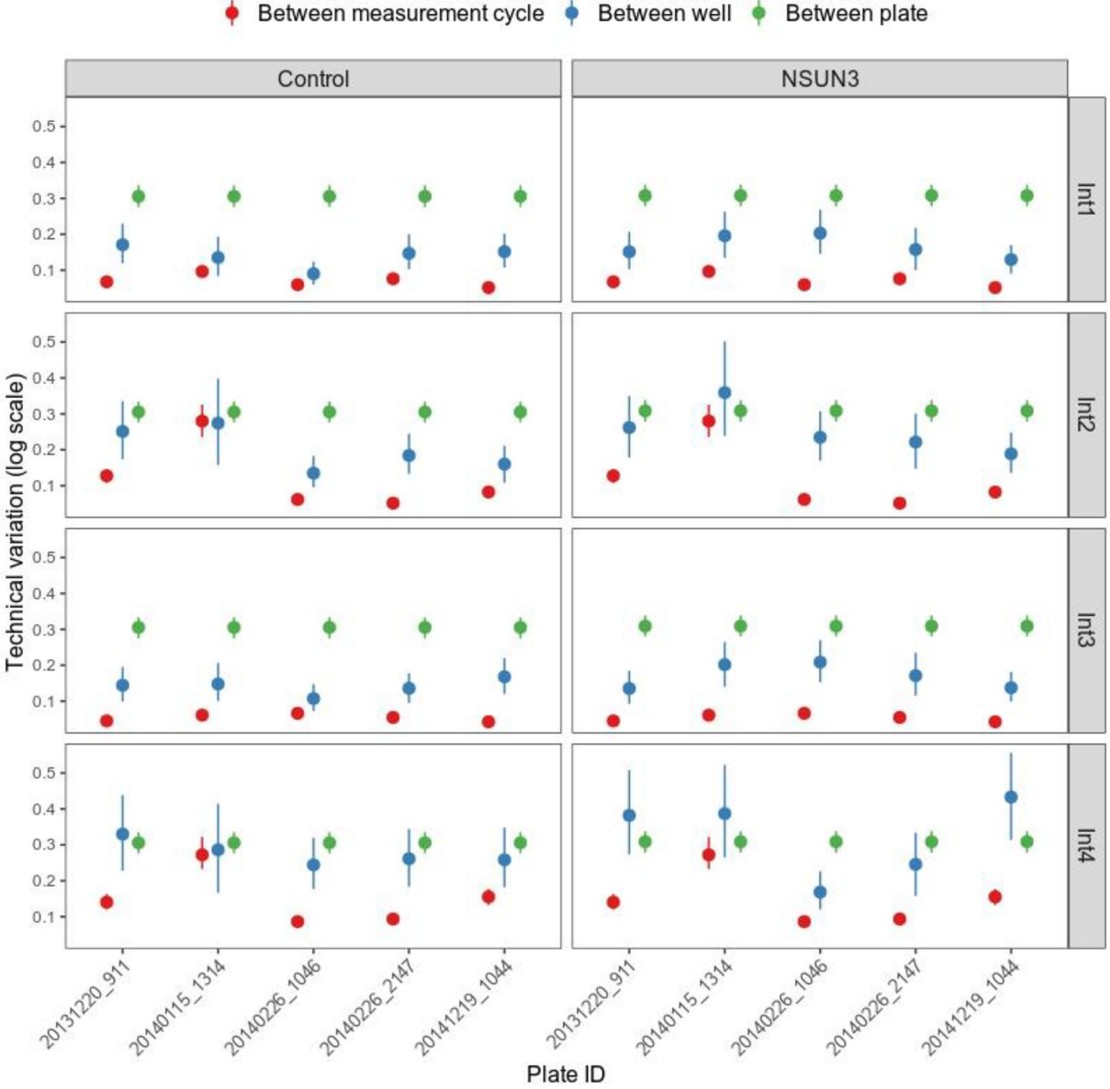

**Fig 2. Technical variations in Seahorse XF OCR data of experiments based on the patient cell line with mutation in NSUN3 gene and the control cell line.** Int1, Int2, Int3 and Int4 represent initial, oligomycin, FCCP and antimycin/rotenone phase. All the variations were on log scale. The dots are the posterior mean and the line segment represent the 95% credible interval.

between plate variation. To visualize the technical variations, we used the cell line with a genetic mutation in *NSUN3* gene (patient cell line 76065). This cell line was measured on five different plates, allowing us to visualize all three technical variations.

OCRbayes calculated the between measurement cycle variation for each plate during each interval. For each cell line in each plate, OCRbayes calculated the between well variation during each interval. Regarding the between plate variation, OCRbayes estimated it for each cell line during each interval. All the variation values were on the log scale. We observed that for both patient and control cell line in the initial phase (Int1) and FCCP phase (Int3), the between plate variation was larger than the between well variation, which itself was larger than the between measurement cycle variations (Fig 2). However, in the oligomycin phase (Int2) and antimycin/rotenone phase (Int4), the between plate variation was not always larger than the other two technical variations (Fig 2). In particular, we observed that the between well

variation was larger than the between measurement cycle and the between plate variation in three plates during the antimycin/rotenone phase (Int4) in the patient cell line.

## Benchmark OCRbayes

To demonstrate that OCRbayes works properly, we applied it to analyze the published Seahorse XF OCR data set containing 26 patient cell lines as well as a control cell line reported by Yepez et al. [13]. This data set contains 6 cell lines that were labeled as positive controls and 2 cell lines that were labeled as negative controls (as explained in the material and methods section).

Based on our analysis, we found 6 patient cell lines that had lower maximal respiration compared to the control cell line with False Discovery Rate (FDR) below 0.05 (Fig 3A and 3B). Among the 6 patient cell lines that showed statistically significant reduction in maximal respiration compared to the control cell line, the patient cell line 73387 (mutation in *PET100* gene) showed the largest effect whereas the patient cell line 76065 (mutation in *NUSN3* gene) showed the smallest effect (Fig 3A). Five of these 6 patient cell lines were labeled as positive controls in the original study [13]. Meanwhile, the two negative controls (patient cell lines 73901 and 91410) had FDR above 0.05 in our analysis.

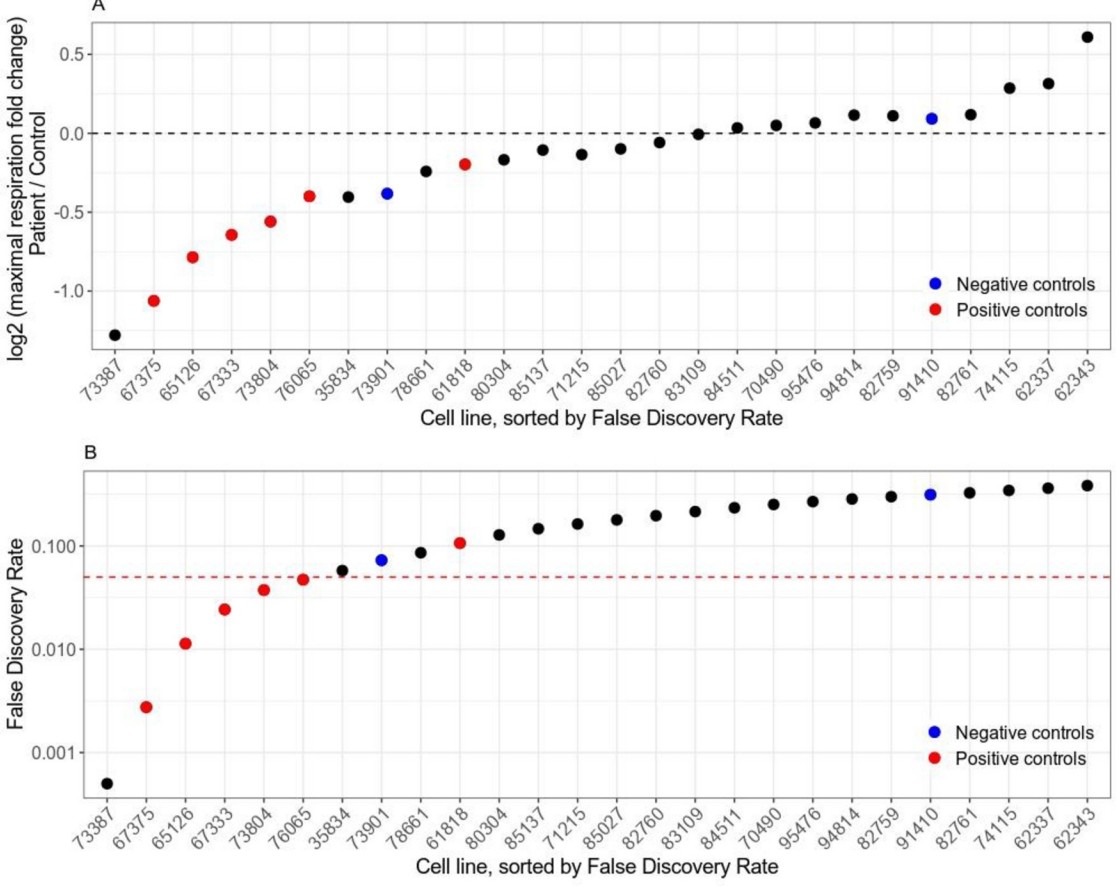

**Fig 3. Maximal respiration change patient vs. control on multiple plates.** (A) average log2 fold change (y-axis) of maximal respiration of all cell lines repeated across plates (x-axis) and their respective controls, sorted by the False Discovery Rate (FDR). Red and blue dots represent positive and negative controls, respectively. (B) similar to (A), but depicting FDR. Red dashed line represents FDR = 0.05.

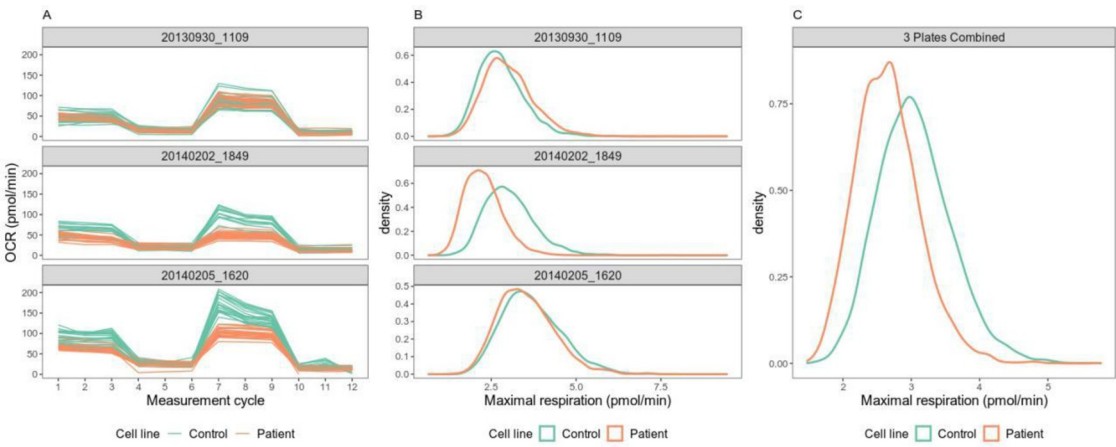

**Fig 4.** (A) Raw OCR values for the cell line with genetic mutation in SFXN4 gene (patient cell line 61818) in three repeated plates. (B) Posterior distributions of maximal respiration of the cell line with genetic mutation in SFXN4 gene (patient cell line 61818) and the control cell line in the same plate. (C) Posterior distributions of maximal respiration of the cell line after combing all three repeated experiments.

Our analysis successfully recalled both negative controls and five out of six positive controls. Interestingly, our analysis showed that there was no significant reduction of maximal respiration in the cell line with a genetic mutation in *SFXN4* gene (patient 61818) that was previously labeled as positive control [13]. This patient cell line was measured on three independent plates on three different days. The range of OCR values of these three plates differed, indicating considerable between plate variation (Fig 4A). The OCR profiles of the patient and control cell line were overlapping in the first plate (Fig 4A). In contrast, the OCR profiles derived from the second and third plate showed OCR decreasing in the patient cell line compared to the control cell line, especially during the FCCP phase (Fig 4A). Meanwhile, we observed considerable variation between the replicate wells as well as measurement cycles in these Seahorse assays (Fig 4A).

In addition, the between measurement cycle variation in the third plate seemed to be larger than the other two plates (Fig 4A). We analyzed these three repeated experiments separately as well as combined them together. Our separate analysis showed that none of the three experiments showed significant reduction in maximal respiration in the *SFXN4* patient cell line (Fig 4B). The posterior mean log2(fold change) and the corresponding 95% credible intervals derived from the first, second and third Seahorse assay were 0.122 [-0.834, 1.04], -0.431 [-1.40, 0.499] and -0.0977 [-1.04, 0.827], respectively. When we used OCRbayes to analyze all the three plates together, we found that the posterior mean log2(fold change) of maximal respiration was -0.200 [-0.877, 0.500] compared to the control cell line (Fig 4C). However, this tendency of reduction in maximal respiration in *SFXN4* patient cell line was not statistically significant.

## OCRbayes can be used to evaluate the probability that a patient fibroblast cell line has an abnormality in mitochondrial respiration

A feature that makes OCRbayes unique from other methods is that OCRbayes can evaluate what is the probability that a patient has abnormality in his or her fibroblast mitochondrial respiration based on a single Seahorse assay or multiple Seahorse assays. To demonstrate this feature, we used two patient cell lines that showed significant reduction in maximal respiration in our analysis. One patient cell line was the *PET100* gene mutation fibroblast (patient cell line

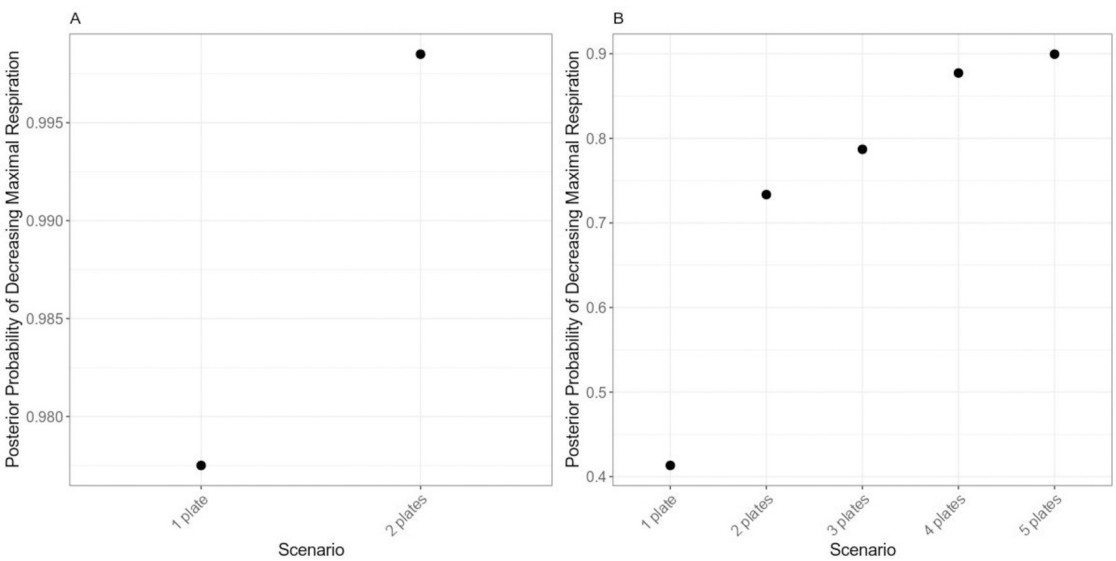

**Fig 5.** (A) Posterior probability of decreasing maximal respiration in patient cell line with mutation in PET100 gene compared to the control cell line in the scenario with 1 or 2 plates. (B) Posterior probability of decreasing maximal respiration in patient cell line with mutation in NUSN3 gene compared to the control cell line in the scenario with 1, 2, 3, 4 or 5 plates.

73387) and the other patient cell line was *NSUN3* gene mutation fibroblast (patient cell line 76065). Among the 6 patient cell lines that showed statistically significant reduction in maximal respiration compared to the control cell line, the *PET100* and *NSUN3* cell line showed the largest and smallest effect, respectively (Fig 3B).

The *PET100* patient cell line was measured on two plates. Based on the first assay, our analysis showed that the posterior probability of this *PET100* mutation carrier having lower maximal respiration than the control was 97.8%. By repeating the experiment once more, the posterior probability increased to 99.8% (Fig 5A). In contrast, the cell line with mutation in *NSUN3* gene (patient cell line 76065) was measured on five different plates. Our analysis showed that based on the first assay, the posterior probability of this *NSUN3* mutation carrier having lower maximal respiration than the control was 41.3%. Repeating the assay once increased the posterior probability from 41.3% to 73.4%. When the assay was repeated for the third, fourth and fifth time, the corresponding posterior probabilities increased to 78.7%, 87.7% and 90.0%, respectively (Fig 5B). In summary, we demonstrate that OCRbayes can be used to evaluate the probability that a patient fibroblast cell line has an abnormality in mitochondrial respiration based on a single Seahorse assay or multiple Seahorse assays.

## Discussion

Although Seahorse XF analyzer is widely used in bioenergetic profiling, its data analysis has not received sufficient attention. A hidden feature of Seahorse XF OCR data is that it contains a complex data structure. The complex data structure is due to the fact that measurement cycles are nested within injections, injections are crossed with wells, and wells are nested in plates. As far as we know, currently there is no data analysis protocol that takes into account this complex data structure, impairing the robustness of Seahorse XF OCR data analysis outcomes. This is because when one ignores the data structure, one also ignores the variations between measurement cycles, between wells and between plates. In order to make the Seahorse data analysis more robust, in this study we developed a Bayesian hierarchical modeling

approach, OCRbayes, which accounts for all these technical variations during the data analysis.

## Seahorse XF OCR measurements are noisy

An OCR value is determined not only by mitochondrial activity, but also by technical noise including 1) between measurement cycle variation, 2) between well variation and 3) between plate variation.

Every phase typically contains three measurement cycles, resulting in three OCR values. Since every measurement cycle starts with a "mix and wait" step to ensure the same baseline of cell values, cell physiology should not substantially change within a phase. Existing tools such as OCR-stats [13] and SHORE [12] use different strategies to select a single data point to represent an injection phase. When different data points were chosen, one can get different outcomes, making the current Seahorse XF data analysis less robust. To avoid this ambiguity, our approach did not select any particular data point, instead modeled all three OCR data. By doing so, we incorporated the uncertainty about "which data point should I choose?" into the data analysis and focused on the average behavior.

The between well variation refers to a common observation that OCR values differ among the replicate wells. One important factor leading to the variation between replicate wells is that cell numbers are not identical in these wells. The wells with more cells would have higher OCR than the wells containing fewer cells. Fortunately, cell number in each well can be quantified experimentally and used for the data analysis [20, 21]. In addition to cell number difference, initial conditions, treatment concentration, or fluorophore sleeve calibration can also contribute to variation between wells. OCRbayes also takes into account the between well variation caused by these unobserved factors.

Between plate variation takes place when the same Seahorse experiment is repeated on different days and on more than one plate. Due to differences in temperature, seeding time, growth time, growth medium, sensor cartridge as well as treatment efficiency, the OCR outcomes will differ between plates [13, 22]. Often the between plate variation is assumed to be the dominant technical variation involved in Seahorse OCR data. Based on our analysis, we showed that this assumption may be appropriate for OCR measurements derived from the initial and FCCP phase, but may not work for the OCR values derived from oligomycin and antimycin/rotenone phase. OCR values in the oligomycin and antimycin/rotenone phase were very small and possibly close to the detection limit of the Seahorse XF in a well. Thus, it is more challenging to accurately measure the OCR in these phases.

## Comparison with other statistical tool for Seahorse XF OCR data analysis

In this work, we compared the maximal respiration in 26 patient cell lines to the control cell line individually as what was done in OCR-stats [13]. Overall our analysis recalled successfully all negative controls and five out of six positive controls. Patient cell line 61818 was labeled as a positive control since this patient was found having a mutation in *SFXN4* gene. A recent study based on erythroleukemic cell line showed that *SFXN4* knockout resulted in significant decrease in all parameters of respiration, including baseline respiration, respiratory ATP synthesis, maximal respiration, and spare respiratory capacity [23]. However, our analysis showed that the maximal respiration of this patient was not significantly different from the control cell line. Our further analysis showed that the patient cell line only showed a tendency of having lower maximal respiration than control in one of the three repeated experiments.

We also noticed that other outcomes were also not identical as what was presented in the OCR-Stats publication [13]. Strikingly, in our analysis *PET100* mutation fibroblast (patient cell

line 73387) showed a significant decrease in maximal respiration compared to the control cell line. However, in the OCR-stats [13], the difference in maximal respiration was not statistically significant in this patient. This patient was diagnosed carrying a homozygous loss of function mutation in the *PET100* gene, which encodes a mitochondrial complex IV biogenesis factor [13, 17, 18]. A homozygous truncating variant (c.142C>T, p.(Gln48*)) in the *PET100* gene was found to lead to a complete loss of enzyme activity, and caused deficiency in complex IV [24]. Therefore, our observation of significant reduction in maximal respiration in *PET100* mutation fibroblast based on the OCRbayes is in line with the observed loss of enzymatic activity in patients carrying the genetic mutations in *PET100*. This difference in analysis outcome highlighted the advantage of OCRbayes which used hierarchical models to incorporate the complex data structure into Seahorse OCR data analysis, helping separate the technical variations from the OCR measurements and identify difference in the biological OCR.

## OCRbayes is potentially used for screening patients with mitochondrial diseases

Since an increase in proton leak or a decrease in basal or maximal respiration are indicators of mitochondrial dysfunction [25], Seahorse XF is potentially to be used for screening mitochondrial disease patients. Besides providing solely statistical significance information as other methods do, OCRbayes allows us to calculate posterior probability that the maximal respiration (or any other respiration metrics) was abnormal in a patient even based on a single Seahorse assay. This feature is also helpful for deciding whether we need to run a single or multiple Seahorse assays for a patient.

For example, the patient cell line with mutation in *PET100* gene showed the largest decrease in maximal respiration in our analysis. The posterior probability for this patient having abnormal mitochondrial respiration after observing a single Seahorse assay was already about 98%. It is reasonable to run just a single assay in this case. In contrast, the cell line with mutation in *NSUN3* gene had the smallest effect size among all the patient cell lines that showed significant reduction in maximal respiration. The posterior probability of this patient having impaired maximum respiration after running a single Seahorse assay was modest (about 40%). Repeating the experiment once more increased the posterior probability from 40% to 70%. Thus in this case, it is beneficial to run multiple Seahorse assays. We think that the posterior probability is a useful metric to help scientists to decide whether to perform extra Seahorse assays on the patient cell lines. The ability of posterior probability calculation by OCRbayes not only allows to make better conclusions about the significance of the effect of the experimental perturbation, it can also prevent from repeating unnecessary, often expensive, experiments. Furthermore, when the availability of patient material or the number of target cells (specific isolated immune cell subsets) is limited, OCRbayes is valuable to exploit the limited data and facilitate proper validation the experiments.

## Strengths and limitations

OCRbayes has several advantages. The first advantage of our approach is that it incorporates various technical variations including between measurement cycle variation, between well variation as well as between plate variation during the estimation of bioenergetic measures. All current methods need to choose a single data point from the three measurement cycles to represent the OCR during a particular phase. This procedure ignores the uncertainty and makes analysis less robust because different choice of data points may lead to different results. The second advantage is that OCRbayes can calculate posterior probability for difference in various bioenergetic measures based on Seahorse OCR data consisting of a single plate or multiple

plates. This is a useful feature for screening samples derived from patients with mitochondrial diseases. A third advantage of OCRbayes is that it can be used for guiding improvement of experimental protocols for running Seahorse assays, because the hierarchical model can explicitly quantify the changes in technical variations resulted from different protocols.

One limitation of this study is that OCRbayes is extensively tested with experimental data derived from human fibroblasts cell lines. This is because our model development was restricted to the limited publicly available Seahorse OCR data sets. To generalize our statistical method to other cell lines than human skin fibroblasts, the prior distributions of those cell lines should be estimated from previously collected Seahorse OCR data. This is because each cell line or cell system has its own growth conditions and metabolic characteristics, making the prior distributions based on the human fibroblasts data not suitable for other cell lines. Furthermore, the prior distributions for plate-to-plate or well-to-well variation are likely dependent on factors such as laboratory and machine. Although many Seahorse studies have been published, few of them provided the raw OCR measurements together with the cell number quantification information. Generalization of our method would benefit from the open availability of raw Seahorse OCR data. This would also facilitate making OCRbayes applicable for other cell lines than human skin fibroblasts.

## Author Contributions

**Conceptualization:** Xiang Zhang, Taolin Yuan, Jaap Keijer, Vincent C. J. de Boer.

**Data curation:** Xiang Zhang.

**Formal analysis:** Xiang Zhang.

**Methodology:** Xiang Zhang.

**Software:** Xiang Zhang.

**Supervision:** Vincent C. J. de Boer.

**Visualization:** Xiang Zhang.

**Writing – original draft:** Xiang Zhang.

**Writing – review & editing:** Xiang Zhang, Taolin Yuan, Jaap Keijer, Vincent C. J. de Boer.

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
