## [Decision Letter · Decision Letter 0]

25 May 2021

PONE-D-21-10491

OCRbayes: A Bayesian hierarchical modeling framework for Seahorse extracellular flux oxygen consumption rate data analysis

PLOS ONE

Dear Dr. Zhang,

Thank you for submitting your manuscript to PLOS ONE. After careful consideration, we feel that it has merit but does not fully meet PLOS ONE’s publication criteria as it currently stands. Therefore, we invite you to submit a revised version of the manuscript that addresses the points raised during the review process.

 Overall, the manuscript has been very well received by the reviewers and could be accepted for publication following addressing all the comments.

We look forward to receiving your revised manuscript.

Kind regards,

Ravirajsinh Jadeja, Ph.D

Academic Editor

PLOS ONE

Journal Requirements:

Reviewers' comments:

Reviewer's Responses to Questions

**Comments to the Author**

1. Is the manuscript technically sound, and do the data support the conclusions?

Reviewer #1: Yes

Reviewer #2: Yes

2. Has the statistical analysis been performed appropriately and rigorously? 

Reviewer #1: Yes

Reviewer #2: Yes

3. Have the authors made all data underlying the findings in their manuscript fully available?

Reviewer #1: Yes

Reviewer #2: Yes

4. Is the manuscript presented in an intelligible fashion and written in standard English?

Reviewer #1: Yes

Reviewer #2: Yes

5. Review Comments to the Author

Reviewer #1: This is an interesting paper which addresses major limitations with the data analysis related to Seahorse oxygen consumption rate assays, especially where the conclusions are drawn from multiple repeats of experiments. The current available analysis methods are limited which consider only a single set of data for each measurement from each plate that doesn’t account for the variability due to other measurements or between the plates assayed on different days. The authors used the Bayesian hierarchical modeling framework as a statistical tool to analyze the OCR data to account for the variables such as between plate, between well and between the measurement cycles. Overall the manuscript is very well written and the experiments are organized to support the hypothesis.

Please address the following

1. Please explain how OCR-stats are different from OCRbayes other than using single measurement in consideration.

2. Selecting a single data point has some rationale. Many times it’s important to choose a single data point as compared to multiple points. Like some times oligomycin takes time to act and its better to take the last measurement. How can you explain that for using OCRbayes?

3. Line 170 states that mitochondrial dysfunction is often reflected in the maximal respiration is not completely true as all other parameters are equally important like proton leak.

4. Is OCRbayes applicable for lab based experiments which are repeated 2-3 times? If not please mention in the limitations.

5. The robustness of the statistical method need to be proved in more than one assay/cells which at least need to be mentioned in limitations.

Reviewer #2: The Seahorse XF-96 analyzer evaluates mitochondrial function in real-time within live cells and would be of great value to test hypotheses on the relevance of mitochondria in various diseases, including cancer, cardiovascular, and neurodegenerative disease pathogenesis.

In this manuscript, Zhang et al. have reported an improved technique called OCRbayes to analyze the Seahorse XF OCR experimental data, which can be used for the diagnosis of patients suffering from mitochondrial diseases. The authors claim that this is a novel protocol that has never been reported elsewhere before.

Here, they have developed a statistically robust method by adopting the Bayesian hierarchical model approach. This approach is adequate to consider all the technical variations which arise due to complex data structures. The OCRbayes is advantageous and more suitable than any other method adopted so far. Also, this approach is ideal for application in the analysis of a patient with mitochondrial abnormalities.

Overall, the methods reported in this manuscript is important for the scientific community studying the mitochondrial function as it is more robust and will allow the investigators to measure mitochondrial function in live cell populations over time, eliminating the need for timed experiments with multiple samples at different time points; thus expediting the research and facilitating explorations of new scientific directions. Moreover, the manuscript is presented in a well-organized fashion, and the methods are well described systematically.

However, the current approach suffers certain limitations as the authors have validated their protocol on only a particular type of cell line and thus requires extended investigation on another type of cell line as mitochondrial dysfunction is linked to various organs, including the heart, brain, kidney, etc. The following comments need to be addressed to improve the current manuscript.

Comments:

Major: None

Minor

1.The author should clarify whether this protocol is applicable to analyze the data generated from other tools such as the Oroboros high-resolution oxygraphy or other similar devices.

.

2a.In their experiments, the authors considered the cell number variations as one of the factors responsible for the in-between well variations in the data within a cell line in a plate. Does minimize this variation by regulating the cell number variations by plating the cells with a multichannel pipette in high accuracy reduces these variations? Considering the following steps may be helpful (a) to mix the cell well before putting it to the reservoir to prevent settling down. (b) Mixing the cells once in every other column/row while plating. (c) Avoiding the outer walls in the experiments since the growth of the cells in these wells are significantly affected by the condensation or air. (d) Further, the slow-growing cells may increase the variability if keeping them in 96 well plates for more than two days.

2b. Is there any variation in their data between adherent and non-adherent cells?.

3. The authors should clarify whether their protocol will apply to the in-vivo settings as in the case of the OCR data generated from the zebrafish embryos in real-time from Simon T bond et al. Method Mol Bio 2018.

4. In this study, the authors have considered data generated from only cell type. The data generated from various sources, as published in Martin P Horan et al. 2012 in the Journal of Gerontology, summarizes multiple data generated from different sources by using the Seahorse XP analyzer. The author should consider discussed data if accessible.

5. Line 47 and 79 are repeated in the introduction/method section and can be rephrased.

6. PLOS authors have the option to publish the peer review history of their article (what does this mean?). If published, this will include your full peer review and any attached files.

Reviewer #1: No

Reviewer #2: No

---

## [Author Response · Author response to Decision Letter 0]

28 May 2021

Dear Dr. Ravirajsinh Jadeja,

Thank you for considering our manuscript for publication in PLOS ONE. We were pleased to read your comments and those of the reviewers. We have addressed the comments of the reviewers point-by-point and revised the manuscript accordingly. They helped us to improve our manuscript and we thank the reviewers for their constructive suggestions.

Reviewer 1

This is an interesting paper which addresses major limitations with the data analysis related to Seahorse oxygen consumption rate assays, especially where the conclusions are drawn from multiple repeats of experiments. The current available analysis methods are limited which consider only a single set of data for each measurement from each plate that does not account for the variability due to other measurements or between the plates assayed on different days. The authors used the Bayesian hierarchical modeling framework as a statistical tool to analyze the OCR data to account for the variables such as between plate, between well and between the measurement cycles. Overall the manuscript is very well written and the experiments are organized to support the hypothesis.

Please address the following

1.Please explain how OCR-stats are different from OCRbayes other than using single measurement in consideration.

Answer: Statistically, OCRbayes is a Bayesian method which is fundamentally different from OCR-stats which is a frequentist tool. In contrast to OCR-stats which builds upon linear regression, OCRbayes applies hierarchical modeling approach and works with the full data set. We explained this now in the introduction section.

“Compared to the currently most advanced Seahorse data analysis tool, OCR-stats, which is a frequentist approach, OCRbayes is based on a fully Bayesian approach due to its flexibility and convenience for constructing the hierachical models.”

2.Selecting a single data point has some rationale. Many times it’s important to choose a single data point as compared to multiple points. Like some times oligomycin takes time to act and its better to take the last measurement. How can you explain that for using OCRbayes?

Answer: We agree with the reviewer that sometimes selecting a single data point can have some rationale, even though that means discarding part of valuable data that could also contain information. However, we also noticed that different labs and tools may use different strategies (one may choose the last data point or others may choose the minimum value) for data selection. In contrast, OCRbayes uses all data as well as prior knowledge about the OCR for a given model system in a particular injection phase, making the Seahorse data analysis more robust. In the case where a drug injection takes time to act, we recommend increasing the “wait” time after injection, so that a stable state is reached before starting a measurement cycle. The dataset we used did have stable states following injections.

3.Line 170 states that mitochondrial dysfunction is often reflected in the maximal respiration is not completely true as all other parameters are equally important like proton leak.

Answer: We agree with the reviewer that besides the maximal respiration, other respiration parameters are also important. We have changed the line 170 as:

“Since the maximal respiration was reported as the primary mitochondrial dysfunction outcome in the Yepez et al. study [13], we reported here the mean log2 fold change of maximal respiration together with the False Discovery Rate (FDR)”

4.Is OCRbayes applicable for lab based experiments which are repeated 2-3 times? If not please mention in the limitations.

Answer: Yes, OCRbayes is applicable for experiments repeated 2-3 times. In fact, in figure 5A we demonstrated that two plates are sufficient to reach statistical significance for the PET100 deficient cell line.

5.The robustness of the statistical method need to be proved in more than one assay/cells which at least need to be mentioned in limitations.

Answer: We agree with the reviewer and we have changed the limitation section as follow:

“One limitation of this study is that OCRbayes is extensively tested with experimental data derived from human fibroblasts cell lines. This is because our model development was restricted to the limited publicly available Seahorse OCR data sets. To generalize our statistical method to other cell lines than human skin fibroblasts, the prior distributions of those cell lines should be estimated from previously collected Seahorse OCR data. This is because each cell line or cell system has its own growth conditions and metabolic characteristics, making the prior distributions based on the human fibroblasts data not suitable for other cell lines. Furthermore, the prior distributions for plate-to-plate or well-to-well variation are likely dependent on factors such as laboratory and machine. Although many Seahorse studies have been published, few of them provided the raw OCR measurements together with the cell number quantification information. Generalization of our method would benefit from the open availability of raw Seahorse OCR data. This would also facilitate making OCRbayes applicable for other cell lines than human skin fibroblasts.”

Reviewer 2

The Seahorse XF-96 analyzer evaluates mitochondrial function in real-time within live cells and would be of great value to test hypotheses on the relevance of mitochondria in various diseases, including cancer, cardiovascular, and neurodegenerative disease pathogenesis.

In this manuscript, Zhang et al. have reported an improved technique called OCRbayes to analyze the Seahorse XF OCR experimental data, which can be used for the diagnosis of patients suffering from mitochondrial diseases. The authors claim that this is a novel protocol that has never been reported elsewhere before.

Here, they have developed a statistically robust method by adopting the Bayesian hierarchical model approach. This approach is adequate to consider all the technical variations which arise due to complex data structures. The OCRbayes is advantageous and more suitable than any other method adopted so far. Also, this approach is ideal for application in the analysis of a patient with mitochondrial abnormalities.

Overall, the methods reported in this manuscript is important for the scientific community studying the mitochondrial function as it is more robust and will allow the investigators to measure mitochondrial function in live cell populations over time, eliminating the need for timed experiments with multiple samples at different time points; thus expediting the research and facilitating explorations of new scientific directions. Moreover, the manuscript is presented in a well-organized fashion, and the methods are well described systematically.

However, the current approach suffers certain limitations as the authors have validated their protocol on only a particular type of cell line and thus requires extended investigation on another type of cell line as mitochondrial dysfunction is linked to various organs, including the heart, brain, kidney, etc. The following comments need to be addressed to improve the current manuscript.

Comments:

Major: None

Minor 

1.The author should clarify whether this protocol is applicable to analyze the data generated from other tools such as the Oroboros high-resolution oxygraphy or other similar devices.

Answer: OCRbayes was designed for OCR measurements generated by Seahorse XF analyzer. At current stage, OCRbayes cannot be directly used to analyze OCR data derived from other technologies such as Oroboros. It would be interesting to follow a similar statistical approach for Oroboros data, but that is beyond the scope of this manuscript.

2a.In their experiments, the authors considered the cell number variations as one of the factors responsible for the in-between well variations in the data within a cell line in a plate. Does minimize this variation by regulating the cell number variations by plating the cells with a multichannel pipette in high accuracy reduces these variations? Considering the following steps may be helpful (a) to mix the cell well before putting it to the reservoir to prevent settling down. (b) Mixing the cells once in every other column/row while plating. (c) Avoiding the outer walls in the experiments since the growth of the cells in these wells are significantly affected by the condensation or air. (d) Further, the slow-growing cells may increase the variability if keeping them in 96 well plates for more than two days.

Answer: We appreciate the input from the reviewer. In a recent study, we have shown that by accurately quantifying cell numbers the variation in the OCR measurements can be reduced [1], other factors like in-plate position could indeed also contribute to well-to-well variation, which is taken into account in the OCRbayes. Since OCRbayes can estimate the variations from the experimental OCR data, we can also use the OCRbayes to optimize the experimental protocol for Seahorse assays. We now included this in the manuscript in line 408:

“A third advantage of OCRbayes is that it can be used for guiding improvement of experimental protocols for running Seahorse assays, because the hierarchical model can explicitly quantify the changes in technical variations resulted from different protocols.”

2b. Is there any variation in their data between adherent and non-adherent cells?.

Answer: we thank the reviewer for the excellent comment. OCRbayes requires prior distributions for all unknown parameters. For adherent and non-adherent cells, we need to estimate their own prior distributions.

3.The authors should clarify whether their protocol will apply to the in-vivo settings as in the case of the OCR data generated from the zebrafish embryos in real-time from Simon T bond et al. Method Mol Bio 2018.

Answer: OCRbayes can be applied to other in vitro or in vivo systems. For each model system, one needs to provide prior distributions for the required parameters. This can be achieved when more experimental data are generated and become accessible.

4.In this study, the authors have considered data generated from only cell type. The data generated from various sources, as published in Martin P Horan et al. 2012 in the Journal of Gerontology, summarizes multiple data generated from different sources by using the Seahorse XP analyzer. The author should consider discussed data if accessible.

Answer: We completely agree with the reviewer on this point. However, the reality is that few publications provided the raw Seahorse data as well as cell number data, required for analysis by OCRbayes. Currently, we are working on setting up a platform for sharing Seahorse raw data. Nevertheless, we have changed the text of our limitation section as follow:

“One limitation of this study is that OCRbayes is extensively tested with experimental data derived from human fibroblasts cell lines. This is because our model development was restricted to the limited publicly available Seahorse OCR data sets. To generalize our statistical method to other cell lines than human skin fibroblasts, the prior distributions of those cell lines should be estimated from previously collected Seahorse OCR data. This is because each cell line or cell system has its own growth conditions and metabolic characteristics, making the prior distributions based on the human fibroblasts data not suitable for other cell lines. Furthermore, the prior distributions for plate-to-plate or well-to-well variation are likely dependent on factors such as laboratory and machine. Although many Seahorse studies have been published, few of them provided the raw OCR measurements together with the cell number quantification information. Generalization of our method would benefit from the open availability of raw Seahorse OCR data. This would also facilitate making OCRbayes applicable for other cell lines than human skin fibroblasts.”

5.Line 47 and 79 are repeated in the introduction/method section and can be rephrased.

Answer: We thank the reviewer for this comment. We have rephrased the line 79 as:

“During a measurement cycle, the Seahorse XF analyzer uses fluorescent oxygen sensors to track OCR.”

Reference

1. Janssen JJE, Lagerwaard B, Bunschoten A, Savelkoul HFJ, Neerven RJJ van, Keijer J, et al. Novel standardized method for extracellular flux analysis of oxidative and glycolytic metabolism in peripheral blood mononuclear cells. Sci Rep. 2021;11: 1662. doi:10.1038/s41598-021-81217-4

---

## [Editor Report · Decision Letter 1]

16 Jun 2021

OCRbayes: A Bayesian hierarchical modeling framework for Seahorse extracellular flux oxygen consumption rate data analysis

PONE-D-21-10491R1

Dear Dr. Zhang,

We’re pleased to inform you that your manuscript has been judged scientifically suitable for publication and will be formally accepted for publication once it meets all outstanding technical requirements.

Kind regards,

Ravirajsinh Jadeja, Ph.D

Academic Editor

PLOS ONE
---

## [Editor Report · Acceptance letter]

2 Jul 2021

PONE-D-21-10491R1 

OCRbayes: A Bayesian hierarchical modeling framework for Seahorse extracellular flux oxygen consumption rate data analysis 

Dear Dr. Zhang:

I'm pleased to inform you that your manuscript has been deemed suitable for publication in PLOS ONE. Congratulations! Your manuscript is now with our production department. 

Kind regards, 

on behalf of

Dr. Ravirajsinh Jadeja 

Academic Editor

PLOS ONE